# A Method for Validating the Structural Completeness of Understory Vegetation Models Captured with 3D Remote Sensing

**Samuel Hillman** [1,2,3,*] **, Luke Wallace** [1] **, Karin Reinke** [1,2] **, Bryan Hally** [1,2] **, Simon Jones** [1,2] **and Daisy S. Saldias** [1]

1   School of Science, RMIT University, Melbourne 3000, Australia; luke.wallace2@rmit.edu.au (L.W.);
    karin.reinke@rmit.edu.au (K.R.); bryan.hally@rmit.edu.au (B.H.); simon.jones@rmit.edu.au (S.J.);
    daisy.san.martin.saldias@student.rmit.edu.au (D.S.S.)
2   Bushfire and Natural Hazards Cooperative Research Centre, East Melbourne 3002, Australia
3   Department of Environment Land Water and Planning, Victorian Government,
    East Melbourne 3002, Australia
*   Correspondence: samuel.hillman@student.rmit.edu.au or samuel.c.hillman@gmail.com; Tel.: +613-9925-9726

**Abstract:** Characteristics describing below canopy vegetation are important for a range of forest ecosystem applications including wildlife habitat, fuel hazard and fire behaviour modelling, understanding forest recovery after disturbance and competition dynamics. Such applications all rely on accurate measures of vegetation structure. Inherent in this is the assumption or ability to demonstrate measurement accuracy. 3D point clouds are being increasingly used to describe vegetated environments, however limited research has been conducted to validate the information content of terrestrial point clouds of understory vegetation. This paper describes the design and use of a field frame to co-register point intercept measurements with point cloud data to act as a validation source. Validation results show high correlation of point matching in forests with understory vegetation elements with large mass and/or surface area, typically consisting of broad leaves, twigs and bark 0.02 m diameter or greater in size (SfM, MCC 0.51–0.66; TLS, MCC 0.37–0.47). In contrast, complex environments with understory vegetation elements with low mass and low surface area showed lower correlations between validation measurements and point clouds (SfM, MCC 0.40 and 0.42; TLS, MCC 0.25 and 0.16). The results of this study demonstrate that the validation frame provides a suitable method for comparing the relative performance of different point cloud generation processes.

**Keywords:** structure from motion; terrestrial laser scanning; validation; 3D remote sensing; vegetation structure; biomass; forest measurement

## 1. Introduction

Development of remote sensing technologies have facilitated the ability to rapidly capture 3D data containing vegetation structural information [1,2]. Accurate measurement of vegetation at different spatial and temporal scales is critical in informing a range of ecological and natural resource management disciplines [3–5]. Estimates of vegetation characteristics such as canopy height, canopy cover, diameter at breast height, and above-ground biomass have been identified as important metrics for carbon accounting, wildlife habitat diversity, precision forestry, fire behaviour modelling and understory forest dynamics for eco-hydrological monitoring [6–10].

Data capture from above the canopy (satellite, aircraft or drone) is the most common approach to characterising the 3D structure of forests [11–14]. Known limitations in observing detailed below

canopy structure from airborne sensors have led to investigation into terrestrial data capture [7,15,16]. The two methods that have been used previously for capturing 3D information describing forests are laser scanning and image-based point clouds [17]. Terrestrial Laser Scanning (TLS) is a ground based-remote sensing technique which utilises the time for a laser pulse emitted from the sensor to reflect off a target and return to the sensor to provide a precise range to that target [18]. Combining this range information with the direction the laser was emitted allows for the 3D location of the target to be determined relative to the sensor [18]. The speed of operation of modern TLS allows for multiple points to be collected rapidly and accurately [19]. TLS point clouds have previously been used in the assessment of below-canopy forest characteristics in particular for examining stem properties; diameter at breast height, stem curve and species and allometry biomass [18–21]. More recently, fine scale vegetation characteristics have also been observed in the context of estimating biomass through allometry [1,16,22,23].

Structure from Motion (SfM) is an image based technique which observes the same features, in multiple images, from different viewpoints to build a point cloud [24]. Prior research has utilised SfM point clouds derived from terrestrial and airborne images to estimate tree-centric characteristics such as canopy height and cover, diameter at breast height, and stem count [14]. Preliminary research using terrestrial SfM point clouds for measuring below canopy vegetation biomass has demonstrated strong correlation to direct measurements, visual assessments, and TLS point clouds in pasture and grassy dry forest environments [1,17,22,25–27]. However, limited research has been conducted to investigate the potential of these point clouds to represent fine-scale vegetation structure between the top intercept height and ground surface.

With prior studies demonstrating potential for terrestrial point clouds for measuring below canopy vegetation, demonstration of the absolute accuracy and correlation to existing methods is seen as an important consideration for adoption on a wider-scale. Validation of point clouds have been largely focused on visual assessment and destructive harvesting to compare biomass estimates [1,16,19,22]. Whilst considered to be the most accurate form of volume assessment, manual collection and weighing is time-consuming, labour intensive and prohibits repeat-sampling [28,29]. Such approaches also only provide an indication of the vegetation volume across the measured area and need to be combined with other measurements to provide an indication of the vertical distribution of this volume [30,31]. Whilst being quick to execute, visual assessments have proven to be subjective and vary substantially between assessors in multiple environmental applications [25,32–35].

Studies that have directly validated the representation of vegetation structure contained within terrestrial point clouds have utilised the point-intercept method [23,28,36–38]. The point-intercept method involves the use of a wooden or metal dowel placed at randomly or regular spaced intervals, with the height and type of vegetation measured at that point. This method of assessment has been used extensively for the measurement of vegetation characteristics in ecological studies [39–42] and shown to be highly correlated with the estimation of plant biomass [43]. A difficulty in using this technique is co-registering the point intercept observations to the point cloud for direct comparison. To achieve this several approaches have been trialled; Bright et al. [38], for example, used markers placed on the ground to establish a coordinate system. Whilst this approach is suitable in areas of low vegetation, accurately locating a point intercept sample in high vegetation cover would be difficult. Loudermilk et al. [28] and Hiers et al. [37] used styrofoam balls to co-register TLS point clouds with manual field measurements. Whilst this may be suitable for volumetric calculations utilising top height, characterisation of the structural content would be challenging to make due to the variation in coordinate systems.

The use of laser pointers have been suggested to provide an alternative to using metal dowels to measure cover especially in studies where the phenomenon of interest is above head height [44–47]. Whilst this technique may provide a less destructive method of measuring vegetation top height due to minimal disturbance of the vegetation, obstruction from vegetation makes the use of lasers to measure vegetation impractical for measuring height of a complex surface and/or near-surface

vegetation structures. The point-intercept method in combination with an appropriate sampling density has demonstrated to be an accurate form of vegetation structure assessment for measuring defined vegetation elements.

The aforementioned studies demonstrate the capability of point cloud capture to provide a representative description of biomass volume relative to destructive sampled weight. What these approaches do not provide is an indication of the quality of the representation of the overall vegetation structure in the point cloud data. Limited research has been conducted in applying methods which directly validate vegetation structural characteristics contained in point clouds.

The primary aim of this manuscript is to develop a method that allows the representation of understory vegetation structure contained within point clouds to be assessed. Two case studies were used to evaluate the performance of the method. The first case study highlighted the effect of the frame on the point cloud. The second case study highlighted how the methodology can be applied at sites that have a range of cover and height attributes. It is shown that the validation frame measurements provide a rigorous assessment for the relative performance of terrestrial point clouds generated using both TLS and SfM technology. The implications of the frame measurements to be used as an indicator for point cloud completeness at multiple stages of the point cloud processing workflow and its potential use across a wider range of environments are also discussed.

## 2. Materials and Method

### 2.1. Validation Frame Design

One of the primary obstacles in the validation of point clouds based on data primitives (such as intercepts derived from the point intercept method) is the co-registration of the two sources of data. To achieve an accurate co-registration of validation and point cloud information, a frame was designed to register point intercept locations and allow point cloud information to be transformed into the same coordinate space.

The frame consisted of a 0.5 m × 0.5 m quadrat on four variable height legs (Figure 1). The size of the frame developed in this study was chosen to align with similar practices in other ecological studies [48–50]. The origin of the coordinate system was the lower-left inner corner of the quadrat frame. The variable height legs allowed the frame to be levelled using a spirit level bubble approximately 1 m from the ground. A sliding rack was used to allow the collection of 8 point intercept measurements simultaneously along the *x*-axis. This rack could be located at eight locations along the *y*-axis, allowing a total of 64 measurement points to be measured at known locations within the coordinate system defined by the frame.

At each measurement point, an aluminum rod of 0.6 cm diameter marked at 1 cm intervals was inserted into the rack to ground level. The rack was designed with two vertical bars spaced 5 cm apart to ensure when inserted the rods remain closely aligned to the *z*-axis with a maximum deviation of 2 cm. For each rod, the method outlined in [44] for taking point intercept measurements can be followed. This involved noting the height above ground, type, and width of any vegetation elements intercepting the rod. The height of the rod at the bottom of the second vertical bar was also observed to determine the location of the ground within the frame's coordinate system.

### 2.2. Point Cloud Collection

The in-field use of the validation frame has been designed to work independently from the point cloud technology (i.e., TLS or SfM). As such, point clouds can be collected in the desired form with the frame placed in the area of observation at the time of capture. Following the point cloud capture, the point intercept reference measurements were taken. As rods are removed to change rack positions, minor disturbances to the vegetation may occur. The adopted best practice was therefore to capture the reference measurements at the end of the survey.

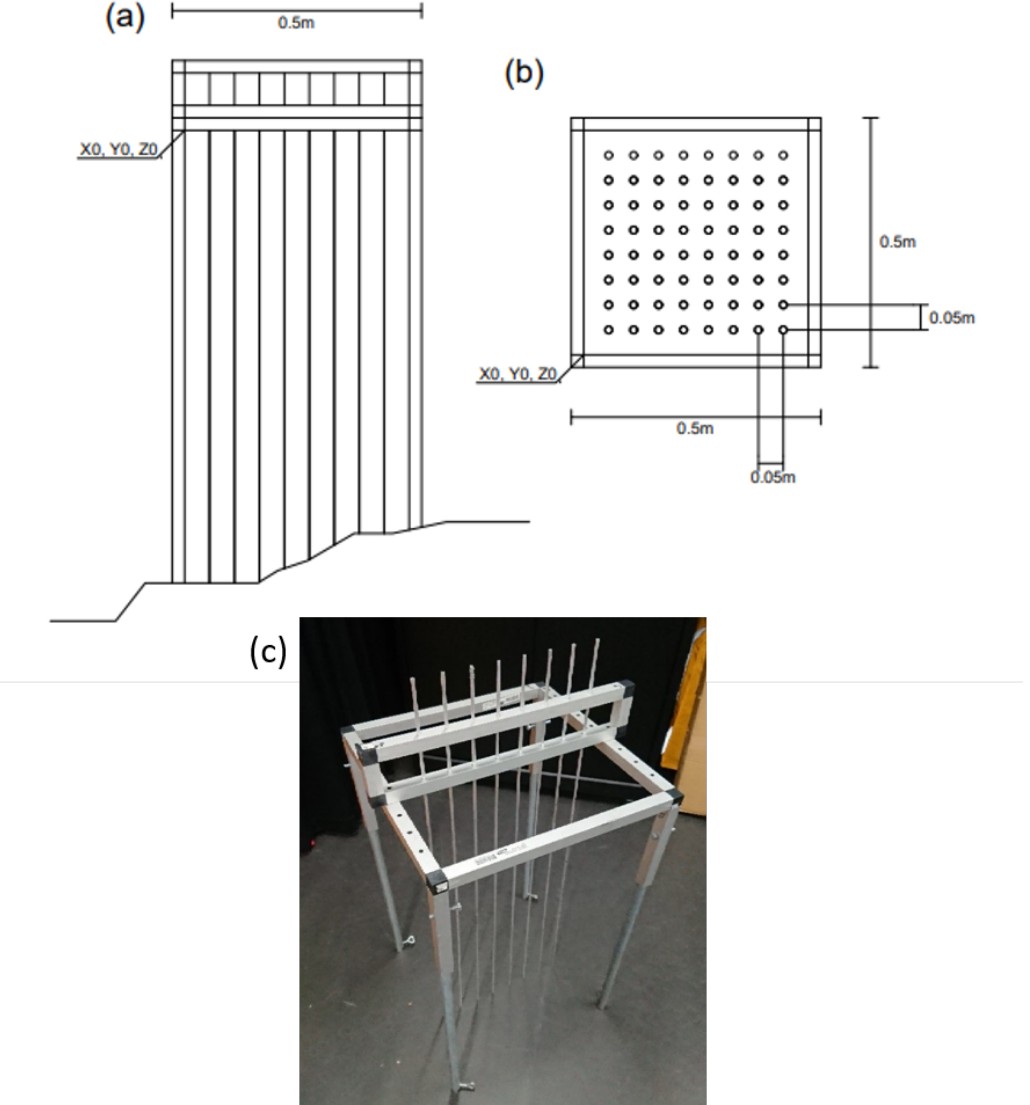

**Figure 1.** (**a**) 0.5 m × 0.5 m validation frame showing the cross-section area sampled using a point-intercept method. Frame was levelled before samples were taken. (**b**) 64 measurement points were taken in a grid spacing of 0.05 m with the spacing kept consistent through the use of a sliding rack. (**c**) Frame setup, demonstrating variable height legs with eight rack positions and eight measurement rods inserted into rack.

### 2.3. Co-Registration of Validation Measurements and Point Clouds

This section describes how SfM and TLS point clouds were co-registered to the validation frame. These two approaches provided an indication as to how point clouds captured using emerging technologies can be co-registered to the frame. For TLS data, scale is an inherent property of the point cloud, thus, in this case, a six-parameter transformation (three translations and three rotations) of the point cloud relative to the frame was defined based on a set of common points. The common points between the two coordinate systems were defined by manual location of the outside edge of the quadrat in the TLS point cloud.

For image based point clouds, point cloud information can be transformed to the frame coordinate system using known locations on the frame as control targets. This is achieved by locating each corner of the frame on four or more images. Once added, the 3D position is estimated in the arbitrary coordinate space using the SfM solution. A seven-parameter transformation (three translations, three rotations and one scale term) is then determined and applied to the point cloud.

### 2.4. Voxelisation

To directly compare the reference data to the point cloud in this study, each dataset was transformed into a binary voxel model indicating vegetation presence and absence. A voxel model for these purposes is a three-dimensional grid domain with the size of the grid cells determining the resolution of the 3D grid [51].

For the reference data, the vertical resolution of the data can be considered to be 10 mm (due to the marks on the rod), while the horizontal resolution was determined to be 20 mm to allow for any slight deviations of the rods from *z*-axis. The conversion of the reference information to voxel space can achieved by indicating if an intersection occurred within a voxel. For the point cloud information, a similar conversion was achieved by counting the number points that fall within the 3D space defined by each voxel, the voxel can then be considered filled if the point count is greater than a count threshold.

### 2.5. Metric Determination and Validation

The validation of the representation of vegetation within the point cloud can be achieved both on the raw data and on secondary metrics (such as vegetation cover and vegetation top height) based on the frame measurements. For validating the raw point cloud information, the presence or absence of voxel data can be directly compared. By determining the number of true and false positive and negatives, the completeness of the point cloud representation can be evaluated. A True Positive (TP) result indicates a point that was present in both the reference measurement and point cloud. A True Negative (TN) result highlights that a point was not present in both the point cloud and reference measurements. A False Negative (FN) indicates the presence of a reference measurement with no corresponding filled voxel. A False Positive (FP) indicates the presence of a filled voxel that does not correspond with a reference measurement. As there is likely to be a higher number of empty voxels in most environments, a Matthew Correlation Coefficient (MCC) was utilised to provide a measure of classification accuracy. This measure has been used previously in the application of point cloud analysis [52,53] to compare observed and predicted binary classifications where the classes are imbalanced [54]. The properties of the intercept can also be examined to determine the type and location of missing information in the point cloud. The error in intercept height can then be summarised using measures of difference such as Root Mean Square Error (RMSE) or Mean Absolute Deviation.

The first step in extracting measurements of a vegetated scene from a point cloud is often the detection of points originating from the ground and the construction of a terrain model. The frame allows the accuracy of such ground definitions derived from point clouds to be assessed by comparing the terrain height at each measurement point to the frame measured height. Following the extraction of the ground, the point cloud can be normalised, which is achieved by subtracting the point's raw height from the ground height at the location of the point. This provides the above-ground height of the vegetation. The accuracy of vegetation height is determined by comparing the normalised height from the point cloud, with a similar measure from the frame data. Point cloud data can also be normalised based on the definition of the ground provided by the frame measurements, allowing the influence of poor ground definition on vegetation metrics to be fully understood. Height metrics such as vegetation top height and vertical profiles can all be validated using this method. Similarly, percentage vegetation cover can be estimated based on the normalised point cloud. For instance, cover can be calculated based on the number of measurement points with an intercept or filled voxel above a height threshold.

### 2.6. Case Studies

Two independent case studies were conducted to assess the effectiveness of the validation frame in capturing structural information on fine scale vegetation. The first case study was designed to assess the impact of the validation frame on the resultant point cloud representation. This was done to

ensure the validation approach did not bias the outcomes achievable from the point cloud technology. The second case study tested the validation frame in a variety of landscapes to assess the effectiveness of the frame at validating terrestrial point clouds in a variety of landscapes. This section first outlines the sensors used in both case studies, before providing details on plot setup, data capture and analysis, and closes with a discussion of the results undertaken within the case study.

### 2.6.1. Sensors

#### TLS Data Collection

A Trimble TX8 scanner set to capture level two quality scans (resulting in a 11.3 mm point resolution at a distance of 30 m) was utilised for both case studies. For Case Study 1, four laser scans were captured approximately 3 m from the frame in order to specifically observe the effects of the frame on point cloud re-construction. For Case Study 2, the plots were scanned using the method outlined by Schaefer et al. [55], which involved collecting a central scan and four orthogonally located exterior scans taken approximately 10 m from the plot centre. Scans took approximately 3 min to capture and were co-registered using Trimble RealWorks 10.1, with automated alignment utilising the inbuilt cloud to cloud matching routine. Manual quality checking was conducted post-registration and adjustments were then undertaken if required.

#### Image Based Data Collection

Images were collected using two consumer grade mirrorless cameras: an Olympus OM-D EM-10 camera and Sony Alpha 6000. Images were captured at approximately 2 m from the reference frame centre with a horizontal spacing or offset between each image of 0.3 m. This resulted in between 30 and 40 images being taken of the area within and around the reference frame. The height of the image capture varied: 1.0 m, 1.4 m and 1.8 m. Six downward looking images over the top of the frame were also captured to complete the image acquisition process.

Images were collated and processed to form a point cloud using Agisoft Photoscan 1.4.0 (Agisoft LLC, Moscow, Russia). Images were aligned to generate a sparse point cloud using the high quality setting. The high quality setting and mild depth filtering was then applied to generate a dense point cloud. Point clouds were then clipped to represent the area of the reference frame with an approximate 1 m buffer.

### 2.6.2. Case Study 1: Effect of Frame on Point Cloud

The purpose of the first case study was to determine whether the validation frame had any influence on the 3D reconstruction of vegetated scene when using either TLS or SfM techniques. To achieve this, scans and images were captured of two plots with understory vegetation in a built environment. The first plot had vegetation of approximately 0.1 m in height and covered 50% of the validation frame area. The second plot represented a more complex environment with the height of the vegetation approximately 0.5 m and covered 80% of the validation frame area. To fully validate the impact of the frame on representing vegetation, images and scans were captured firstly without the frame, then with the frame present, and finally with the frame removed again. Care was taken when placing and removing the frame to reduce disturbance to the vegetation inside the plot.

#### Point Cloud Co-Registration

When the frame was placed in the scene, point clouds were co-registered and transformed into voxel space using the approach outlined in Section 2.3. Without the frame in the scene, discrete features were identified in the already registered point clouds and used to provide the correct registration. To achieve this, the plots were carefully selected in built environments to provide an abundance discrete targets (brick corners, stairs and seat edges) identifiable within both the TLS and SfM point clouds.

Analysis

For assessing the impact on the presence of the validation frame on the point cloud, the area enclosed by the frame was extracted from point clouds captured both with and without the frame present. An assessment on the point cloud completeness was then conducted by comparing the point cloud with frame in place with the two point clouds without the frame. The analysis method outlined in Section 2.5 was then followed; however, in this case, the point cloud with the frame from each technology was used as the reference source. This allowed the differences in the point clouds with and without the frame to be quantified.

2.6.3. Case Study 2: Validation

The purpose of the second case study was to implement the validation methodology developed in this paper, and to assess the effectiveness of SfM and TLS derived point clouds in detecting vegetation components at a fine scale.

The study was undertaken within Australian forest landscapes. Sites located in Australia were chosen to provide a range of cover and structural characteristics for assessment in southeastern Australia.

Site Descriptions

The validation frame was tested in multiple environments tested to examine the frame's ability to measure vegetation structure. The High Camp study site is situated in a heath dry forest near Kilmore, Victoria, Australia, approximately 80 km north of Melbourne (Figure 2). The overstory largely consisted of low, open eucalypt forest with an open crown cover. The understory consisted of sparse ericoid-leaved shrubs including heaths and peas. There was minimal surface litter and a range of exposed rock sections creating variable terrain.

Royal Park is an urban parkland environment located in Parkville, Vic, Australia, 3 km north of Melbourne's CBD (Figure 2). The overstory vegetation consisted of sparse Eucalypt trees, approximately 15 m tall. The understory consisted of a surface layer of coarse woody chips (e.g., tan bark) with some larger woody debris. Near-surface vegetation consisted of native shrubs with complex leaf arrangement.

The Silvan1 site is located in moist forest near Monbulk, Vic, Australia, approximately 40 km to the east of Melbourne (Figure 2). This plot was located less than 50 m from Silvan Reservoir. There was also evidence of a planned burn occurring in the previous 5–10 years. The overstory largely consisted of medium to tall forests dominated by *Eucalyptus obliqua* and *Eucalyptus radiata*. The understory vegetation was dominated by native shrubs. The surface litter was measured at approximately 25 mm in height and consisted mostly of leaf litter with some woody debris. Near-surface vegetation was dominated by native shrubs.

The Silvan2 site is similarly located to Silvan1, in moist forest near Monbulk, VIC, Australia (Figure 2). This plot was located at a higher elevation in comparison to the Silvan1 plot with no evidence of previous planned burn events. Increase in elevation along with a change in aspect produced varied vegetation to Silvan1. The overstory largely consisted of medium to tall forest dominated by *Eucalyptus obliqua* and *Eucalyptus radiata*. Understory vegetation was dense, with a high proportion of Aristia species. Surface litter was measured at approximately 25 mm and consisted mostly of leaf litter with some woody debris. Near-surface vegetation was dominated by native shrubs and aristida.

The Ridgeway site was located in dry sclerophyll forest near Hobart, Tasmania, Australia (Figure 2). Evidence of a planned burn event (completed in 2015/2016) season was evident throughout the plot. The overstory of the plot consisted of *Eucalyptus pulchella* trees of varying age and ranging in height from 4.7 to 16.2 m. The understory of the plot consisted of low to medium height (0.5–2 m) shrubs and grasses. Surface litter mainly consisted of leafs and woody debris.

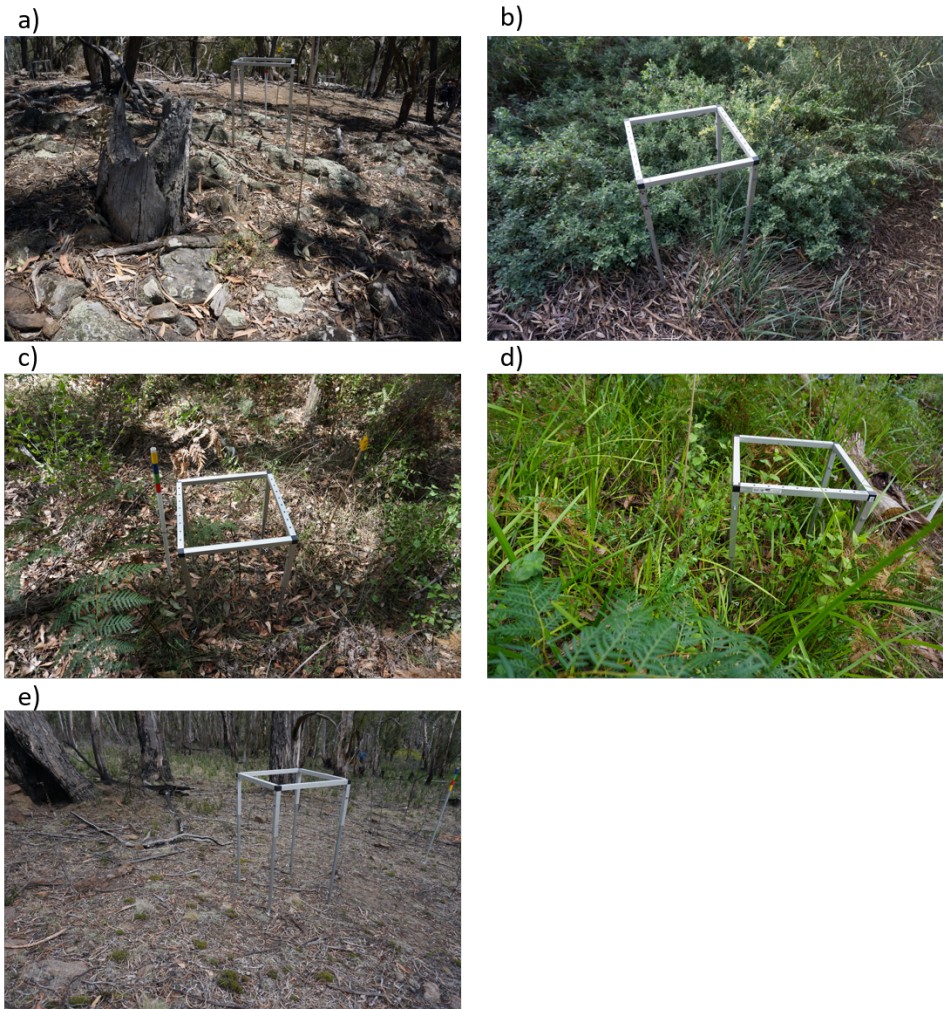

**Figure 2.** Validation frame was tested in a variety of landscapes ranging in height and cover: (**a**) High Camp, Victoria, Australia; (**b**) Royal Park, Victoria, Australia; (**c**) Silvan1, Victoria, Australia; (**d**) Silvan2, Victoria, Australia; and (**e**) Ridgeway, Tasmania, Australia.

Plot Set-Out

For all sites, at least one circular plot of 10 m radius was established. The centre of each plot was marked with a star picket or wooden stake. For each plot, two transects were established in opposing orthogonal directions. Along each transect, two validation frames were randomly located using random numbers chosen within 0–9 m and 11–20 m.

Point Cloud Filtering

Point clouds generated using both SfM and TLS had filtering algorithms applied to reduce noise beneath the ground in the case of SfM, and to remove spurious points in the point cloud produced from the TLS scans [19,26].

Point Cloud Normalisation

Ground points were identified in both TLS and SfM point clouds using the Cloth Simulation Filter [56]. The settings for extracting the ground were optimised to obtain a set of parameters that would produce a ground surface across all plots and minimise RMSE against reference ground measurements (Table 1). A 0.02 m DTM was generated by constructing a Triangular Irregular Network (TIN) from the identified ground points. The height of the TIN facet at the centre of each cell is

attributed to the DTM. TIN interpolation was used to ensure each point intercept sample had an associated ground height estimate. Next, the point cloud was normalised based on each point's height above the TIN. This normalisation procedure provided a representation of the point cloud in relation to the ground.

**Table 1.** Filter settings applied to the Cloth Simulation Filter for SfM and TLS point clouds to identify ground points in all sites. The same parameters were used for generating point clouds for each technology source.

| Setting | SfM Value | TLS Value |
|---|---|---|
| Cloth Resolution (m) | 0.05 | 0.01 |
| Class Threshold (m) | 0.04 | 0.03 |
| Rigidity | 3 | 3 |
| Time Step | 0.50 | 0.20 |
| Iterations | 1000 | 1000 |

Data Analysis

To assess the performance of TLS and SfM in comparison to the reference measurements, the point clouds were assessed for their ability to represent the first intercept (highest vegetation intercept). This assessment of point cloud completeness was undertaken on non-normalised point clouds to mitigate the influence of errors introduced due to the ground height estimates. Subsequently, all intercepts measured during the validation process were compared to the respective point clouds. Vegetation height and cover were calculated from both the reference measurement and derived voxel spaces. Cover is defined as the proportion of the validation frame area that has vegetation above 10 cm. In the context of the validation frame, this is defined as the number of rods for which one or more intercept was recorded above 10 cm divided by 64. Difference in cover were summarised as a single difference value for each frame, while differences in first intercept height within each frame were summarised using RMSE calculated as follows:

$$RMSE = \sqrt{\frac{1}{n} \sum_{i=1}^{n} \left( \frac{f_i - pc_i}{\sigma_i} \right)^2} \tag{1}$$

where $f_i$ is the first intercept (or ground) measured from the frame and $pc_i$ is the first intercept (or ground height) at the same location measured using the point cloud.

An assessment of the TLS and SfM point cloud's ability to measure the ground was also made and summarised using RMSE. Assessments of cover and ground were completed in absolute terms, and on the normalised voxel space. This allowed for identification of vegetation elements (absolute) and the effect of determining a ground surface in altering the vegetation representation in the point cloud (normalised).

## 3. Results

### 3.1. Case Study 1

The TLS point clouds had a strong agreement between point clouds captured with and without the frame present. The low cover site had an MCC 0.89 and 0.87 and high cover site had agreement of MCC 0.82 and 0.84. Similar results were achieved with point clouds derived from SfM with little effect observed from the validation frame (low cover observing MCC 0.92 and 0.93, high cover observing MCC 0.73 and 0.77) (Figure 4). This indicates the validation frame had minimal impact on the utility of both TLS and SfM point clouds in representing vegetation.

Figures 3 and 4 show that any differences in the point clouds typically occurred within or on the edge of vegetation elements for both technologies. This suggests that the majority of the differences between the point clouds were due to other effects and not the presence of the frame. Differences in

data capture geometry, windy conditions causing vegetation to move during data collection and small co-registration errors are likely causes of inaccurate vegetation reconstruction.

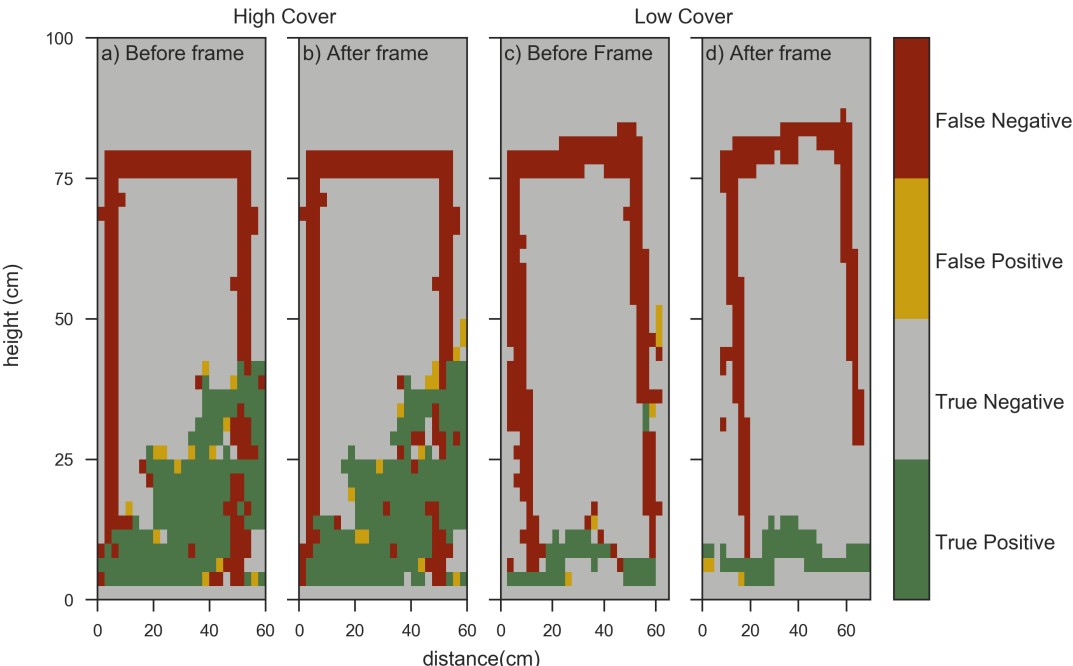

**Figure 3.** Visualisations of the agreement between point cloud data captured at high cover and low cover plots using TLS workflows. The differences are shown between point clouds with and without the frame in place: (**a**,**c**) the comparison to the point cloud captured before the frame was introduced; and (**b**,**d**) the comparison after to the point cloud captured after the frame was removed.

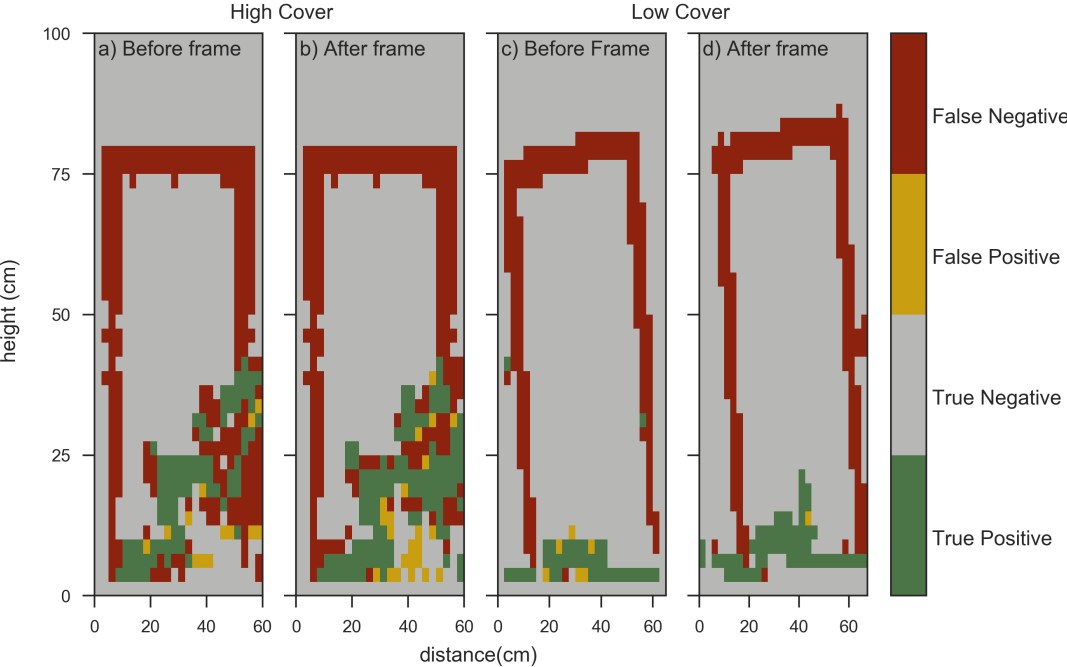

**Figure 4.** Visualisations of the agreement between point cloud data captured at high cover and low cover plots using SfM workflows. The differences are shown between point clouds with and without the frame in place: (**a**,**c**) the comparison to the point cloud captured before the frame was introduced; and (**b**,**d**) the comparison after to the point cloud captured after the frame was removed.

### 3.2. Case Study 2

#### 3.2.1. Correlation of Vegetation Structure

Comparison of SfM and TLS non-normalised point clouds to validation measurements demonstrate that both techniques capture representative vegetation structure. All sites displayed positive correlation with reference data (Tables 2 and 3). Results suggest that SfM point clouds had a stronger correlation with the reference data than TLS point clouds (Table 2). Further, in environments with clearly defined vegetation (>0.02 m diameter), correlation between reference data and SfM and TLS point clouds was moderate (Royal Park: SfM, MCC 0.51, and TLS, MCC 0.37; and Ridgeway: SfM, MCC 0.66, and TLS, MCC 0.47). Reduced distance to the area of interest (less than 1.5 m) is predicted to be a major factor in the improved correlation seen in the SfM point clouds. In complex environments with fine vegetation (<0.02 m), there is lower correlation between reference measurements and point clouds from both SfM and TLS (Silvan1 and Silvan2: SfM, MCC 0.40 and 0.42, and TLS, MCC 0.25 and 0.16). This is due to the reduced likelihood of small structural elements being captured within the point cloud due to their physical size. Lower correlation of cover and biomass estimates has also been seen in other studies with fine-scale vegetation [1,22].

**Table 2.** Summary of confusion matrix results between SfM Non-normalised point cloud and validation frame data. For each plot, True Positives (TP), True Negatives (TN), False Positives (FP), False Negatives (FN) and Matthews Correlation Coefficient (MCC) are given.

| Plot Name | TP (%) | TN (%) | FP (%) | FN (%) | MCC |
|---|---|---|---|---|---|
| High Camp | 333 (2.67) | 11,658 (93.41) | 311 (2.49) | 178 (1.43) | 0.56 |
| Royal Park | 237 (6.50) | 3053 (83.69) | 178 (4.88) | 180 (4.93) | 0.51 |
| Silvan1 | 568 (4.25) | 11,453 (85.62) | 531 (3.97) | 824 (6.16) | 0.40 |
| Silvan2 | 557 (3.27) | 15,148 (88.98) | 491 (2.88) | 828 (4.86) | 0.42 |
| Ridgeway | 575 (4.05) | 13,091 (92.14) | 307 (2.16) | 235 (1.65) | 0.66 |

**Table 3.** Summary of confusion matrix results between TLS Non-normalised point cloud and validation frame data. For each plot, True Positives (TP), True Negatives (TN), False Positives (FP), False Negatives (FN) and Matthews Correlation Coefficient (MCC) are given.

| Plot Name | TP (%) | TN (%) | FP (%) | FN (%) | MCC |
|---|---|---|---|---|---|
| High Camp | 265 (0.98) | 25,856 (95.28) | 599 (2.21) | 416 (1.53) | 0.33 |
| Royal Park | 289 (5.02) | 4735 (82.20) | 371 (6.44) | 365 (6.34) | 0.37 |
| Silvan1 | 385 (1.20) | 29,380 (91.81) | 499 (1.56) | 1736 (5.43) | 0.25 |
| Silvan2 | 232 (0.44) | 49,759 (95.05) | 549 (1.05) | 1812 (3.46) | 0.16 |
| Ridgeway | 623 (1.59) | 37,190 (94.95) | 923 (2.36) | 432 (1.10) | 0.47 |

#### 3.2.2. Accuracy of Ground Estimation

TLS and SfM point clouds from all sites show a greater amount of error in ground estimates as vegetation cover above 0.1 m increases (Figure 5). This pattern is evident in both SfM and TLS point clouds at High Camp (sparse cover with low height of vegetation) with a low RMSE (0.01–0.03 m). SfM and TLS point clouds with measured rod cover greater than 40 per cent have a higher RMSE for ground estimates (0.05–0.09 m). Consistently, high cover vegetation sites (such as Silvan1, Silvan2 and Royal Park) have deep leaf litter layers, which cause an underestimation of the height of the ground. Generally, TLS point clouds are more accurately able to estimate the ground surface. This is due to the greater penetration of the laser through to the surface of the ground [22]. SfM point clouds are also predicted to have lower correlation due to the ground filter settings applied to the SfM point clouds which over-estimated the height of the ground. Depth filtering algorithms applied during the point cloud construction process for SfM also reduces the potential for detailed ground re-construction

in complex environments. These filter settings could be fine-tuned for each site, however this was deemed out of scope for this paper.

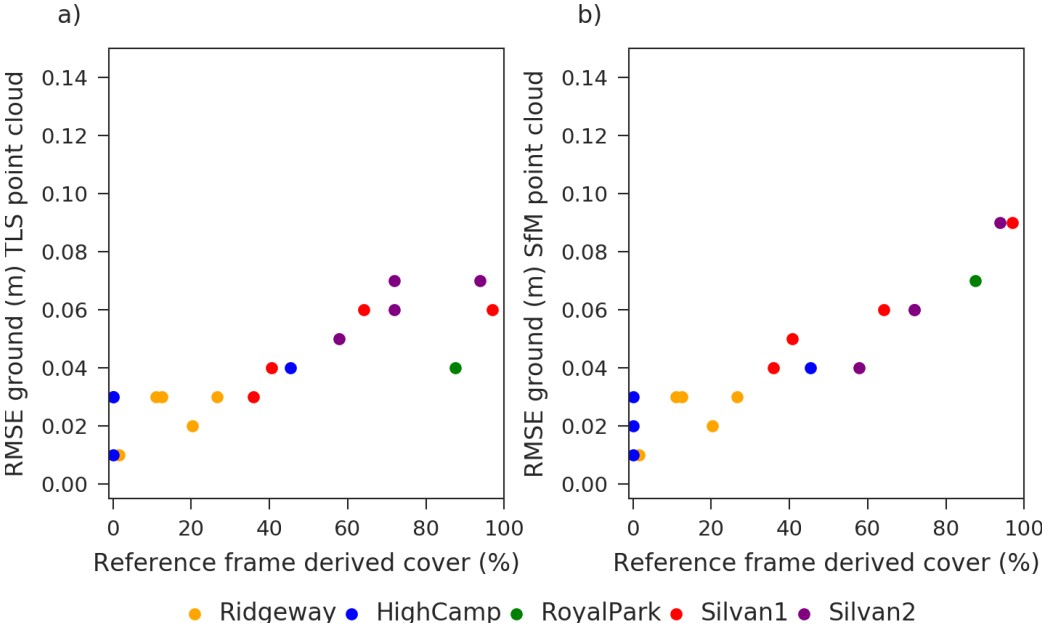

**Figure 5.** TLS (**a**); and SfM (**b**) point cloud RMSE of ground surface compared to cover derived from reference frame measurements. The error in the surface of the ground relative to cover percentage at each measurement point is demonstrated. Each site had multiple plots—for example, High Camp with four sites.

### 3.2.3. Normalised Correlation between Point Clouds and Reference Measurements

Normalising SfM and TLS point clouds, with subsequent removal of the ground intercept from both the reference measurements and point clouds, resulted in lower correlation to reference measurements (Tables 4 and 5). This was prevalent at all sites and most evident at sites with complex vertical arrangement of vegetation (Silvan1 and Silvan2). These results are attributed in both point clouds to the ground filter settings which had included vegetation that was on, or above the ground, in the ground estimate. This observation highlights one of the drawbacks to using remote sensing techniques in deep litter bed environments and supports the integration of direct measurements such as biomass harvesting with remote sensing technologies in the future. Similar to non-normalised point clouds, sites that had vegetation with small structural elements show poor correlation to reference measurements. For both TLS and SfM, low MCC correlations were observed for High Camp from the normalised point clouds. The derived ground surface was overestimated in height due to the rocky features at this site that were not excluded from the ground surface. Fine twigs of less than 0.05 m present at one of the sites in High Camp were also difficult to reconstruct, further reducing the correlation seen at this site.

**Table 4.** Summary of confusion matrix results between SfM normalised point cloud and validation frame data. For each plot, True Positives (TP), True Negatives (TN), False Positives (FP), False Negatives (FN) and Matthews Correlation Coefficient (MCC) are given.

| Plot Name | TP (%) | TN (%) | FP (%) | FN (%) | MCC |
|---|---|---|---|---|---|
| High Camp | 41 (0.25) | 15,717 (96.67) | 116 (0.71) | 384 (2.36) | 0.15 |
| Royal Park | 217 (3.20) | 5955 (87.78) | 239 (3.52) | 373 (5.50) | 0.37 |
| Silvan1 | 423 (1.79) | 21,231 (89.66) | 584 (2.47) | 1442 (6.09) | 0.27 |
| Silvan2 | 337 (1.10) | 28,381 (92.77) | 423 (1.38) | 1451 (4.74) | 0.26 |
| Ridgeway | 84 (0.35) | 23,092 (96.34) | 142 (0.59) | 651 (2.72) | 0.19 |

**Table 5.** Summary of confusion matrix results between TLS normalised point cloud and validation frame data. For each plot, True Positives (TP), True Negatives (TN), False Positives (FP), False Negatives (FN) and Matthews Correlation Coefficient (MCC) are given.

| Plot Name | TP (%) | TN (%) | FP (%) | FN (%) | MCC |
|---|---|---|---|---|---|
| High Camp | 46 (0.50) | 8666 (94.63) | 133 (1.45) | 313 (3.42) | 0.16 |
| Royal Park | 185 (6.57) | 2239 (79.51) | 151 (5.36) | 241 (8.56) | 0.41 |
| Silvan1 | 261 (1.79) | 13,000 (89.09) | 241 (1.65) | 1090 (7.47) | 0.28 |
| Silvan2 | 163 (0.66) | 23,149 (93.46) | 251 (1.01) | 1205 (4.87) | 0.19 |
| Ridgeway | 236 (1.44) | 15,500 (94.70) | 236 (1.44) | 395 (2.41) | 0.41 |

### 3.2.4. Validation of First Intercept Height

Moderate correlation were noted for both SfM and TLS point clouds to reference measurements of first intercept height at HighCamp, Royal Park, Silvan1 and Ridgeway (Figures 6 and 7). SfM point clouds were not successful at resolving fine scale vegetation present at the top of the point cloud, for example Silvan1 and Silvan2 sites where a poor correlation to manually measured heights ($r^2$ 0.32 and RMSE 0.18 m) was recorded. Small amounts of wind are likely to be a contribution to this poor correlation. Similar work completed by Wallace et al. [1] and Cooper et al. [22] also highlighted poor correlations of image based point clouds to destructive sampling of biomass in complex vertical structure and fine vegetation (e.g., woodland environments and grasslands). In this study, the sampling approach for SfM point clouds was not predicted to be a major contributor to error as the assessor captures images in close proximity (3 m) around the site of interest. TLS estimates of first intercept height had slightly stronger correlation to manually measured heights in complex environments. Occlusion from vegetation between the frame and the scanner locations as well as wind effects and movement in vegetation between scans may contribute to a lower correlation in TLS scans. The sampling approach used in this case study meant that scans could be approximately 5 m away from the sample site, which may contribute to a loss of detail.

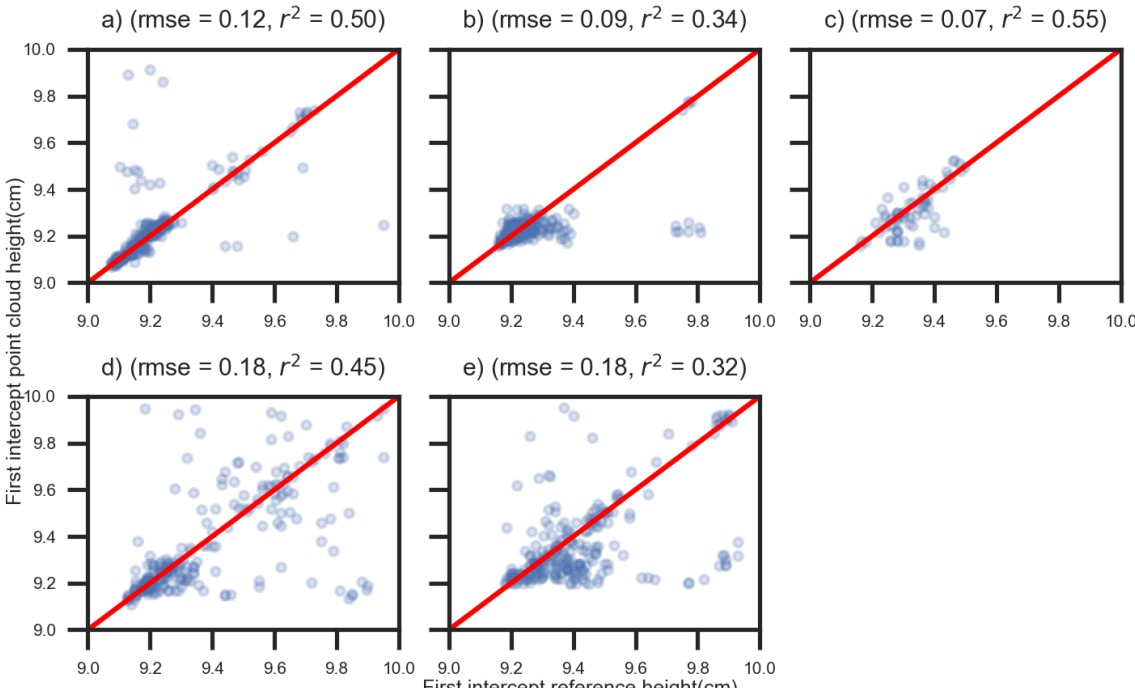

**Figure 6.** SfM estimate of first intercept height derived from point clouds in comparison to direct measurements at: (**a**) High Camp; (**b**) Ridgeway; (**c**) Royal park; (**d**) Silvan1; and (**e**) Silvan2. Both the *x* and *y* axes show height relative to the origin of the reference frame.

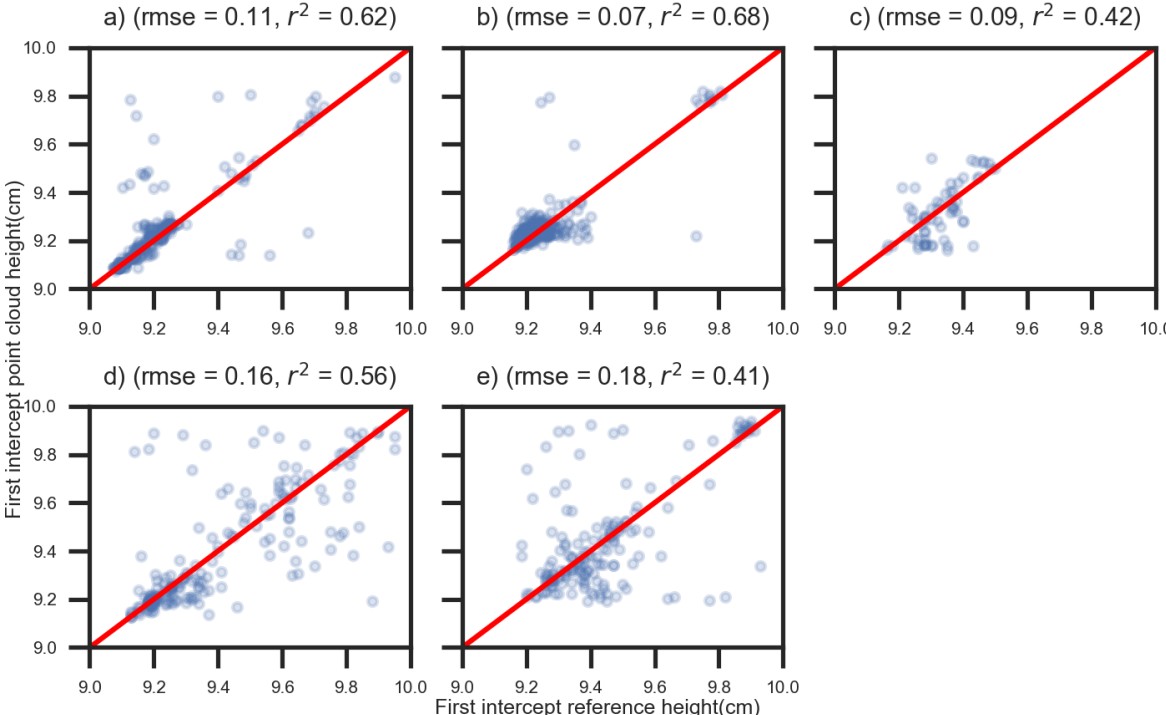

**Figure 7.** TLS estimate of first intercept height derived from point clouds in comparison to direct measurements at: (**a**) High Camp; (**b**) Ridgeway; (**c**) Royal park; (**d**) Silvan1; and (**e**) Silvan2. Both the *x* and *y* axes show height relative to the origin of the reference frame.

### 3.2.5. Validation of Vegetation Cover

The estimates of cover from TLS and SfM point clouds are generally underestimated in comparison to the measurements of cover derived from the validation frame. Greater agreement is seen in environments where there is relatively sparse vegetation (High Camp and Ridgeway) (Figure 8). Investigation of SfM and TLS point clouds in complex vegetation environments (Royal Park and Silvan) highlighted little fine-scale vegetation being reconstructed and therefore underestimating cover. This should however be balanced against the fine scale nature of this vegetation and the capacity for this vegetation element to be over represented in the frame estimate of cover given the resolution of reference frame measurements is 10 mm and the vegetation element is less than 5 mm.

Improved cover estimates are seen when the reference ground surface is utilised (instead of derived point cloud ground) in both SfM and TLS point clouds (Figure 9). SfM point clouds generated in complex vegetation sites (Silvan2) where the ground surface is calculated from reference measurements improves the accuracy of the cover estimates from a difference of $-34.05\%$ to $1.18\%$ (reference cover measurement is 93%). This improvement is observed to be due to the correction of the ground surface with vegetation previously excluded from cover analysis now included. Similarly, in TLS point clouds, estimates of cover improve in complex vegetation types when using the reference measurements of the ground.

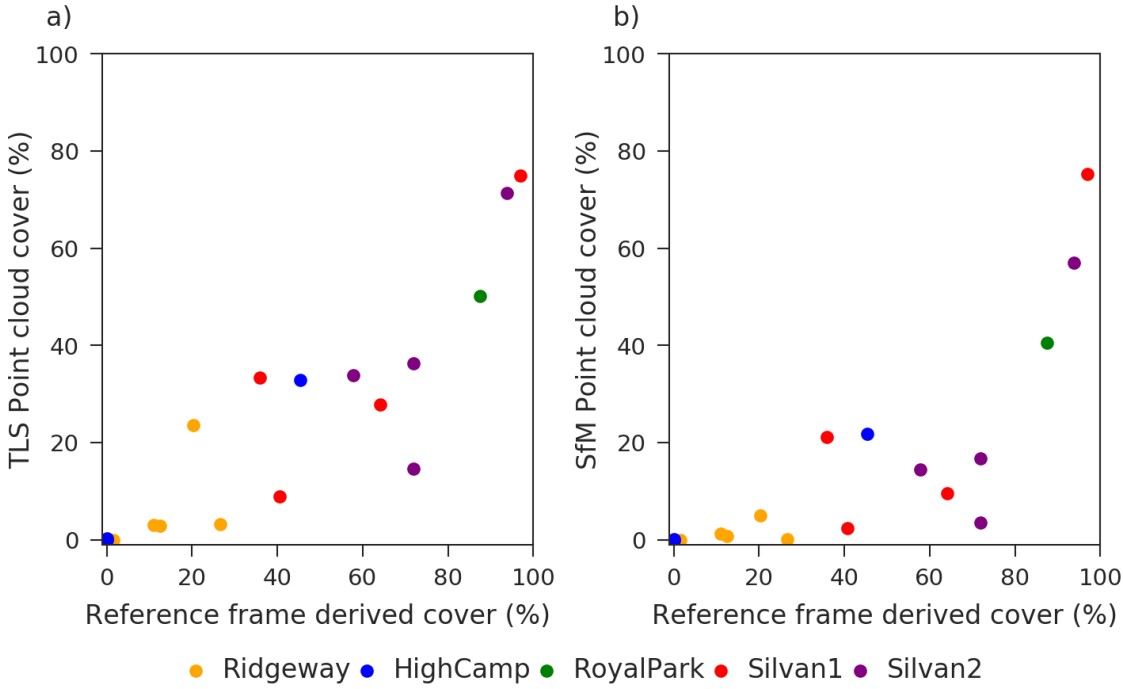

**Figure 8.** TLS (**a**); and SfM (**b**) point cloud cover compared to cover derived from reference frame measurements.

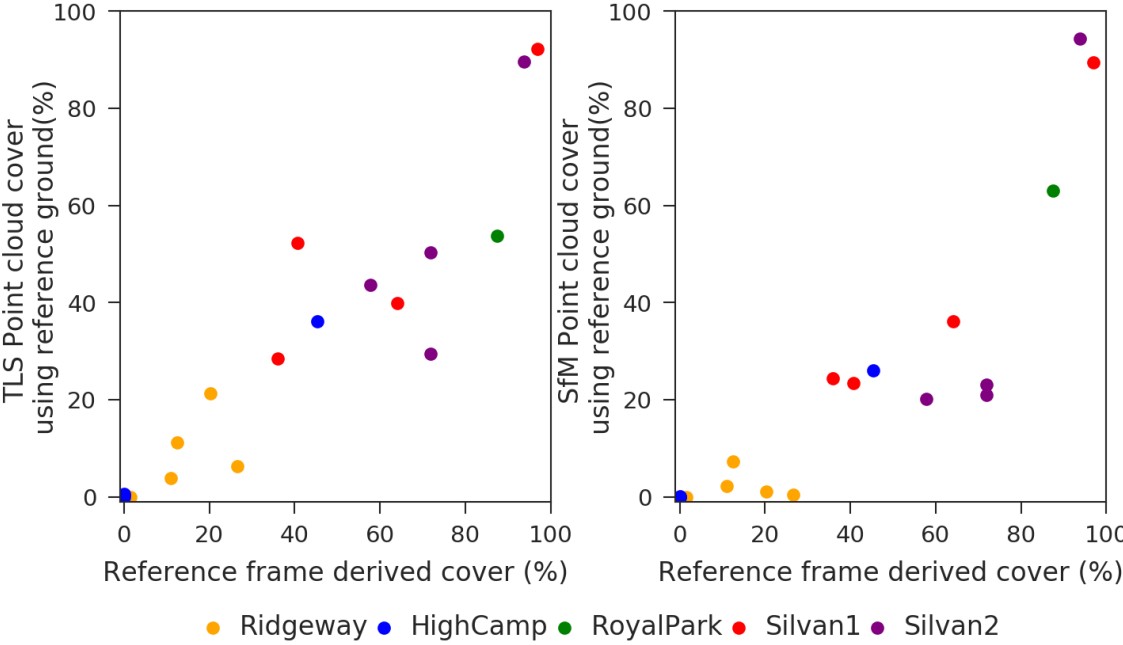

**Figure 9.** TLS (**a**); and SfM (**b**) point cloud cover derived using reference ground measurements compared to cover derived from reference frame measurements.

## 4. Discussion

The validation frame developed in this research provides a portable, robust device to intensively assess the accuracy of point clouds for describing vegetation structure. The method provides a rigorous means for validating the information content within point clouds. Previously, validation approaches for point clouds have not been able to directly compare the representation provided by 3D remote sensing technology without utilising a indirect measure, such as biomass, mean height or 2D cover [1,19,22,28].

The method presented here allows validation of the data through the full structure of the point cloud itself, not just the top height of vegetation. The measurements from the validation frame can be used at varying stages of the point-cloud processing workflow. The second case study presented here highlights that validation with the frame provides insight into the vegetation structure present while also allowing errors in other metrics to be quantified. For example, the effect of different ground filters can be assessed for the most accurate estimation. Previously, the effect of high frequency changes in the terrain has not been explored from these types of point clouds. The frame allows the terrain to be validated at 5 cm intervals and the effect of error in subsequent measurements or models (for example, vegetation height, litter bed depth or biomass) to be evaluated. This differentiates the approach from previous studies, which have only been able to hypothesise as to the causes of the unexplained variance when validating using biomass or visual estimates [22,26,27].

The validation frame can provide insight into potential errors in ecological metric calculations. For example, missing information as seen in sites with complex vertical structure (Silvan1 and Silvan2) was demonstrated to be due to wind-driven errors in fine vegetation with small mass, and by occlusion from vegetation in the frame or surrounding vegetation. This information detected in the validation process then allows the determination of the importance of the missing information in relation to the desired metric calculation. This extra information may allow alternative processing strategies, such as the extraction of metrics that are not reliant on the ground to be used or corrections to be applied to the estimate of some metrics [57].

The frame and method developed provides a non-destructive approach for validating point clouds. The results from Case Study 1 show that the environment being sampled has minimal change with the addition of the validation frame. This is in contrast to direct forms of assessment such as destructive sampling of biomass, which removes all vegetation and prohibits repeat sampling and change detection to occur. There is greater certainty in the alignment of the reference measurements with the point cloud can as the image capture and laser scans are conducted with the validation frame in-situ.

Current visual assessments do not provide an accurate, objective quantification of vegetation structure [25,32,33]. The validation frame establishes a coordinate system allowing direct analysis of point clouds and the information content below the first intercept, including the vegetation structure and ground measurement. To achieve this, care must be taken when positioning the vertical rods to ensure minimal disturbance of the vegetation. This process was difficult in environments with fine scale vegetation (less than 0.01 m) which were susceptible to wind and also minor disturbances from the measurement rods. An alternative sampling approach using a vertically sliding frame and data collection to record presence and absence of vegetation within 10 cm × 10 cm × 10 cm voxels with biomass harvesting could be utilised to allow for some movement of vegetation through wind at the trade-off of precision [58]. Additionally, the time taken to complete a full assessment of the frame area including the harvesting and subsequent drying of biomass is expected to be more time consuming than the approach outlined in this research.

In this study, we chose a frame size that is commensurate with recent studies utilising biomass harvesting as a source of information [48–50]. Nevertheless, the simple frame design allows the approach to be adaptable to different scales and resolutions (or density of horizontal measurements). For larger areas, distributing these frames throughout the captured area would allow the potential uncertainty in understory vegetation measurements from point clouds to be better understood in that environment. In such examples, the number of frames and their placement would be dependent on the vegetation community being sampled, the point cloud sensor sampling approach and the area being captured. This allows the approach to be applied beyond the environments presented in this paper and adapted depending upon the purpose of validation and metric generation.

## 5. Conclusions

New methods for creating 3D representations using localised remote sensing are improving our ability to reproduce and evaluate vegetation structure. These methods, primarily using SfM and TLS,

generate point clouds which are information rich and enable comprehensive quantitative descriptions of vegetation. These methods are becoming common practice and in some instances replacing traditional visual assessments and other indirect forms of characterising the natural environment. However, the accuracy and completeness of the reproductions of vegetation structure using these methods is untested. This paper describes an innovative solution that addresses the challenge of validating the information content of terrestrial point clouds of vegetation. The proposed method is non-destructive and provides a reference system accurately linking the validation information and point cloud data. Through two case studies, it was shown that a number of sources of unexplained variance in both TLS and SfM point cloud vegetation models can be described using the developed approach.

**Author Contributions:** Conceptualisation, S.H., L.W., K.R. and S.J.; Data curation, S.H., L.W., K.R., B.H., S.J. and D.S.S.; Formal analysis, S.H. and L.W.; Investigation, S.H. and L.W.; Methodology, S.H., L.W., K.R. and S.J.; Project administration, K.R. and S.J.; Supervision, L.W., K.R. and S.J.; Validation, S.H.; Visualisation, S.H.; Writing—original draft, S.H.; and Writing—review and editing, S.H., L.W., K.R., B.H., S.J. and D.S.S.

**Funding:** The support of the Commonwealth of Australia through the Bushfire and Natural Hazards Cooperative Research Centre program is gratefully acknowledged.

**Acknowledgments:** The support of the Commonwealth of Australia through the Bushfire and Natural Hazards Cooperative Research Centre and the Australian Postgraduate Award is acknowledged. Chathura Wickramashinge, Ritu Taneja and Ahmad Fallatah are acknowledged for assistance in the collection of the data. Melbourne Water, and in particular Tim Saunders, are gratefully acknowledged for providing access to the Silvan sites.

**Conflicts of Interest:** The authors declare no conflict of interest. The founding sponsors had no role in the design of the study; in the collection, analyses, or interpretation of data; in the writing of the manuscript, and in the decision to publish the results.

## Abbreviations

The following abbreviations are used in this manuscript:

| | |
|---|---|
| SfM | Structure from Motion |
| TLS | Terrestrial Laser Scanning |
| RMSE | root mean squared error |
| MCC | Matthews correlation coefficient |
| TP | true positive |
| TN | true negative |
| FP | false positive |
| FN | false negative |
| TIN | triangulated irregular network |
| DTM | digital terrain model |

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
