# Peer review of "A Method for Validating the Structural Completeness of Understory Vegetation Models Captured with 3D Remote Sensing"

_remotesensing, doi:10.3390/rs11182118_

Round 1
Reviewer 1 Report
This study develops a method for validating understory vegetation structure reconstruction via ground based remote sensing techniques.
The authors provide a concept that could better help quantify the accuracy of ground based remotely sensed estimates of various forest structure variables.
The major shortfall I see in this study is that the authors do not provide any "validation" of their validation frame. That is to say, if the end goal is to quantify how well remote sensing methods represent the real world by comparing it to the model of the world derived by the frame used in this study, then some other metric must also be given to quantify the uncertainty associated with the frame. For example, by measuring the exact same spot with slightly different configurations of the frame and assessing the RMSE and confusion could give us a better understanding of the robustness of this technique. Further, the authors should provide the error associated with the frame ground cover estimates with another independent method such as manual image segmentation. Even though the frame provides a more manual approach to quantifying structure it is still developing a model which must also be quantified if this is to be used for validation purposes.
Further, to show the increased ability to assess structure using this technique, the authors should also show how this method compares against other validation methods on the same variables. This current study does not describe any increase in the level of validation accuracy when quantifying common validation variables using other techniques.
Minor comments include:
1) More proof reading needs to be done. Long, confusing sentences and interesting use of punctuation (such as the use heavy use of parentheses) is prevalent.
2) Provide a reference for RMSE metrics. For example, display values as % of mean or std.
3) Be careful about using editorial comments such as on line 380 or using the term significant in a non statistical way.
4) The pictures of the frame should be in the main body, not in the appendix. Also, your pictures don't show all of the rods discussed in this article. Why not?
Author Response
Thank you for your comments, we appreciate the feedback on the manuscript. Please see attachment for response to comments.
A document highlighting the changes to the manuscript is also supplied.
Kind regards,
Samuel Hillman, Luke Wallace, Karin Reinke, Bryan Hally, Simon Jones and Daisy S Saldias

Reviewer 2 Report
I find the work of great interest in landscape monitoring
3d analysis is in great evolution and can provide useful indications also in the regeneration of the landscape
I believe that we need to encourage the development of these studies to make them more and more efficient and simple in the application to make them usable even to environmental consultants who are not experts in remote sensing.
Author Response
Thank you for your comments, we appreciate the feedback on the manuscript. A document highlighting the changes to the manuscript is also supplied.
Kind regards,
Samuel Hillman, Luke Wallace, Karin Reinke, Bryan Hally, Simon Jones and Daisy S Saldias
Reviewer 3 Report
This manuscript describes an innovative and unique approach developed to validate the accuracy of vegetation classification derived from 3D point clouds generated using SfM and TLS. The paper is well-written and organized clearly. The methodology developed may have practical applications and usefulness as an approach to evaluate accuracy of field versus technology-derived vegetation classification.
2.1 – Validation frame design – how was the scale/size of the frame determined? Is it based on typically collected point clout/TLS data? What drove selection of the dimensions? Are rods used to obtain samples of vegetation? Explain with more detail how samples are taken. Refer reader to the appendix for pictures of the frame and include a specific picture of the frame (not just study sites) in Appendix.
Section 2.2 could be combined with the previous section since it does not actually describe point cloud collection. This is described more clearly in sections 3.1.1 and 3.1.2.
Figure 2: the approach described in the caption could also be clearly described in the text. It is a bit difficult to follow. Figure appears before it is first referenced.
Section 3.3 – minor – rationale for using a TIN to derive cell-center elevation? Why not use the elevations of the points to determine the elevation selected for each grid cells? The raw points are used to generate a TIN which is a more coarse representation of the elevation structure. Include a microscale DEM for visualization?
Tables 2 and 3 – raw values for TP/TN/FP and FN -- better to provide % (normalize by total number points/cells)? Or provide N.
Describe vegetation types found in each site for case study 2 in greater detail. It makes sense that ground elevation may be difficult to quantify in different vegetation types. How does scale of data improve this understanding?
RMSE is suddenly mentioned on 4.2.2. Explain how RMSE was calculated, what values were used, N etc. Not clear how it can range (.01 to .03). Figures 3 is confusing. Perhaps consider a bar graph?
Figures 4 and 5 are confusing. Is this an accuracy assessment of elevation? What does each data point represent?
Discussion - Tighten link between what you found and biomass estimations. Discuss scale. How does scale of validation frame relate to typical studies? Recommend how this method could be applied for future researchers wishing to apply this methodology to calculate accuracy statistics in similar studies.
Typos - Line 284 – change “there “ to “their”, line 353 “an” indirect line 7 – “a” field
Author Response
Thank you for your comments, we appreciate the feedback on the manuscript. A document highlighting the changes to the manuscript is also supplied.
Please see attachment for response to comments.
Kind regards,
Samuel Hillman, Luke Wallace, Karin Reinke, Bryan Hally, Simon Jones and Daisy S Saldias

Reviewer 4 Report
General comments: Accurate characterization of 3D forest structure is important for understanding forest post-disturbance recovery, forest fire behavior and so on. Using the point-intercept method, this study attempt to evaluate the performance of point clouds technology (TLS, SfM) in capturing the 3D structure of forests. The authors have performed an impressive analysis with different cases. However, several points are needed to be addressed before publication.
One major issue is that the method of this study is mainly designed for understory vegetation rather than the whole forest ecosystem. This point has been highlighted by Lines 76-77. However, the title of the manuscript as well as the abstract seem that the method in this study was developed for 3D structure of the whole forest ecosystem.
Please provide more details about the research site, including vegetation characteristics, climate information, soil texture, etc. This could be important for the readers from other forest regions to evaluate whether this method is effective in their ecosystems.
It's also unclear why 0.5*0.5 was chosen for the frame size. Did the authors tested other sizes? Are the correlations reported by Fig. 4-5 strongly affected by the size of frame?
The introduction needs more work. First of all, line 46-47, how to understand ‘However, ……potential of two point cloud methods to represent the fine-scale vegetation structure’. ‘Fine-scale’ refers to ‘structure within point clouds’? Here, the authors introduce the two approaches capturing the 3D forest structure (line 26-46), and then illustrate the shortcomings of validating the point clouds with visual assessment and harvesting (line 48-47). It seems that the ‘validate the vegetation structure contained within the cloud point’ (line 74-75) is the motivation of this study? If yes, please highlight this motivation and the significance in assessing the structure within cloud point. The last paragraph is a bit simple. I suggest to clarify your objectives /hypotheses. The current version only outlines the rest sections of the manuscript (Methods, Cases, Discussion…..).
Please re-organize the Methods part. For example, please add one section to clarify the sites descriptions, the rational for choice of sites.
The results section also needs re-organization. The authors have stated the aims of those two cases (line164-169). However, the current subtitle ‘4.1 Case Study 1’ and ‘4.2 Case Study 2’ is difficult to understand. I wonder whether the findings support the objectives listed in the Introduction. I suggest to improve the results, and test the objectives listed in the introduction. In this way, this manuscript could be more coherent among the results and introduction.
For the Discussion part, please clearly emphasize the advantages of this validation frame, and discuss the possible limitations. Please make a concise summarization of the findings in the Conclusion.
Author Response

(The authors gave the same response as above.)
